# Gastroesophageal reflux GWAS identifies risk loci that also associate with subsequent severe esophageal diseases

Jiyuan An[1], Puya Gharahkhani [1], Matthew H. Law [1], Jue-Sheng Ong [1], Xikun Han [1], Catherine M. Olsen [2], Rachel E. Neale[3,4,5], John Lai[6], Tom L. Vaughan[7], Ines Gockel[8], René Thieme[8], Anne C. Böhmer [9,10], Janusz Jankowski[11], Rebecca C. Fitzgerald [12], Johannes Schumacher[9,10,13], Claire Palles[14], BEACON, 23andMe Research Team, David C. Whiteman[2] & Stuart MacGregor [1]

Gastroesophageal reflux disease (GERD) is caused by gastric acid entering the esophagus. GERD has high prevalence and is the major risk factor for Barrett's esophagus (BE) and esophageal adenocarcinoma (EA). We conduct a large GERD GWAS meta-analysis (80,265 cases, 305,011 controls), identifying 25 independent genome-wide significant loci for GERD. Several of the implicated genes are existing or putative drug targets. Loci discovery is greatest with a broad GERD definition (including cases defined by self-report or medication data). Further, 91% of the GERD risk-increasing alleles also increase BE and/or EA risk, greatly expanding gene discovery for these traits. Our results map genes for GERD and related traits and uncover potential new drug targets for these conditions.

[1] Statistical Genetics, QIMR Berghofer Medical Research Institute, Brisbane, QLD, Australia. [2] Cancer Control, QIMR Berghofer Medical Research Institute, Brisbane, QLD, Australia. [3] Cancer Aetiology and Prevention, QIMR Berghofer Medical Research Institute, Brisbane, QLD, Australia. [4] School of Public Health, The University of Queensland, Brisbane, QLD, Australia. [5] School of Public Health and Social Work, the Queensland University of Technology, Brisbane, QLD, Australia. [6] Centre for Epidemiology and Biostatistics, The University of Melbourne, Melbourne, VIC, Australia. [7] Division of Public Health Sciences, Fred Hutchinson Cancer Research Center, Seattle, WA, USA. [8] Department of Visceral, Transplant, Thoracic and Vascular Surgery, University Hospital of Leipzig, Leipzig, Germany. [9] Institute of Human Genetics, University of Bonn, School of Medicine & University Hospital Bonn, Bonn, Germany. [10] Department of Genomics, Life & Brain Center, University of Bonn, Bonn, Germany. [11] Royal College of Surgeons in Ireland, Dublin, Ireland. [12] Medical Research Council (MRC) Cancer Unit, Hutchison-MRC Research Centre and University of Cambridge, Cambridge, UK. [13] Center for Human Genetics, University Hospital of Marburg, Marburg, Germany. [14] Institute of Cancer and Genomic Sciences, University of Birmingham, Birmingham, UK. A full list of consortium members appears at the end of the paper. Correspondence and requests for materials should be addressed to S.M. (email: stuart.macgregor@qimrberghofer.edu.au)

Esophageal adenocarcinoma (EA) is a fatal cancer with a high mortality rate[1]. Barrett's esophagus (BE) is a precancerous conversion of the normal stratified squamous epithelium of the distal esophagus to columnar epithelium[2]. Gastroesophageal reflux disease (GERD), the frequent regurgitation of stomach acid and bile, is the main risk factor for both BE and EA[3–6].

GERD has a significant socioeconomic burden due to its chronic nature and high prevalence, with approximately 20% of the population affected in western countries[7]. Expenditure on GERD is enormous ($15–20 billion in the US alone in 2006[8]), with spending chiefly on medications. Medications that aim to alleviate or reduce stomach acid secretion, include antacids, histamine–receptor antagonists, and proton–pump inhibitors[9]. However, the efficacy of these medications varies considerably, and most people need prolonged or lifelong use. Furthermore, some have resistance to these medications and, in some cases, medication is insufficient and surgical interventions are required[9]. Developing a better understanding of the etiology of GERD may lead to improved management strategies, such as development of novel or repurposed treatments, ultimately reducing the incidence of BE and EA.

Previous twin studies have shown a significant genetic contribution to the etiology of GERD, with an estimated heritability of 30–40%[10,11]. We recently showed that GERD has a polygenic basis, and estimated a high genetic correlation between GERD and BE ($r_g = 0.77$, SE = 0.24), and between GERD and EA ($r_g = 0.88$, SE = 0.25)[12]. Thus in addition to improving our understanding of GERD, identifying genetic variants for GERD will likely inform our understanding of the genetics of BE/EA. However, previous work[13] has not identified any genome-wide significant ($P < 5 \times 10^{-8}$) risk loci for GERD.

In this study, we perform a large genome-wide association study (GWAS) meta-analysis of GERD, using population-based studies from the UK, USA, and Australia. We aim to: (1) validate the use of self-reported reflux medication as a proxy for GERD in GWAS studies in order to increase statistical power; (2) identify risk loci for GERD; (3) investigate the effect of GERD risk loci on BE and EA; (4) identify the extent of genetic overlap between GERD and its known risk factors (e.g., body mass index (BMI)) as well as other complex traits; and (5) find candidate drugs that target significant genes.

## Results

**GWAS of GERD.** We first undertook five GERD GWASs using three GERD-related data sets from the UK biobank (UKBB) study (ICD10, self-reported GERD, and use of GERD medication), the QSkin study (heartburn and GERD medication use from Pharmaceutical Benefits Scheme (PBS) medical records), and from self-report GERD from the 23andMe data set. Given the differences in phenotype definition across cohorts, we assessed the similarity of the genetic effects across the cohorts by estimating the LD-score genetic correlation ($r_g$) between them. The $r_g$ values were close to 1 in all cases (Fig. 1), except for QSkin where the sample size was too small to allow reliable estimation of genome-wide $r_g$[14]. For all datasets (including QSkin), the correlation between the logarithmic scale odds ratios (log ORs) of the peak single nucleotide polymorphisms (SNPs) was also high (Supplementary Data 1). The strong genetic correlations across the GWAS results justified a meta-analysis of these data sets (UKBB where the three phenotype definitions were first combined and run as one analysis to build the largest nonoverlapping case–control set, 23andMe, and QSkin).

GERD is known to be strongly correlated with BE and EA[12]; this was confirmed in this study by estimating the genetic correlation between GERD (from the above meta-analysis) and a combined BE and EA dataset. The combined data set comprised 13,792 cases and 31,211 controls (Fig. 1), from a meta-analysis of UKBB data (EA and BE cases vs. controls) and independent cohorts from a previously published study[15] (cohorts from Barrett's and Esophageal Adenocarcinoma Consortium [BEACON], Cambridge, Oxford, and Bonn). The GERD-EA/BE genetic correlation was 0.47 (SE = 0.03).

Using an estimated GERD prevalence of 12% among Europeans[16], we calculated the GERD SNP heritability ($h^2$) on the liability scale as 11.3% (SE = 0.004) from the combined GERD GWAS meta-analysis (altering the specified prevalence

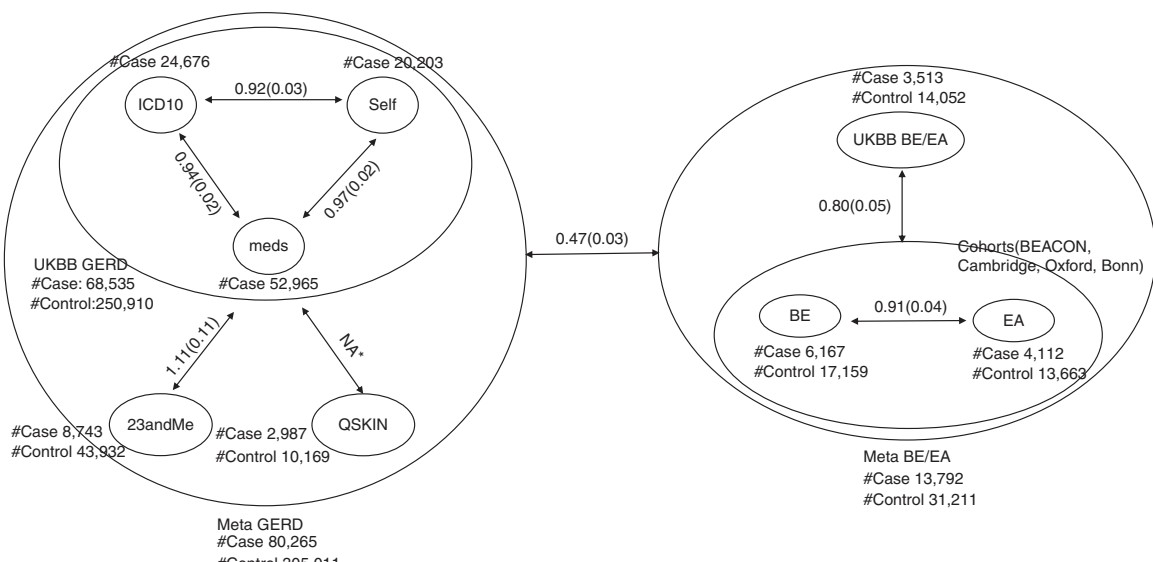

**Fig. 1** Genetic correlation between phenotypes. The lines with two arrows show the genetic correlation (standard error in brackets) from the result of LD-score regression $r_g$, genetic correlation estimates from the LD score regression. There is sample overlap between UKBB cases with either ICD10, self-report or medicine based GERD and the numbers do not add up to the total UKBB samples size. As the correlation is computed as the estimated covariance divided by product of the estimated standard error of the two traits, the correlation estimates may be slightly >1 when the correlation is high. NA* sample size is too small to estimate $r_g$

does not change the SNP heritability appreciably—for example, at a prevalence of 25% $h^2$ only changes to 14.3%). The LD-score intercept for this combined analysis was 1.04 (SE = 0.008), indicating no strong evidence for inflation due to population structure or sample overlap[14]. Defining statistically independent SNPs based on a conditional approach in GCTA[17] (see Methods), 25 SNPs were associated with GERD in our meta-analysis, representing 25 statistically independent associations (Tables 1 and 2). A Manhattan plot of the GERD GWAS meta-analysis is shown in Fig. 2, with the Quantile–Quantile plot (QQ plot) shown in Supplementary Fig. 2.

Gene-based tests (Supplementary Data 5) were conducted using the MAGMA[18] software based on the per-SNP GWAS summary results for GERD. We identified 49 genes that are associated with GERD after Bonferroni correction for 19,427 genes tested ($P < 2.57 \times 10^{-6}$; although many were found by per-SNP tests, 20 were only significant using gene-based tests (Table 3). We also conducted analysis using MetaXcan[19], a gene-based approach that uses gene expression derived from the GTEx project data and the association summary statistics from the GERD GWAS to test the association between genes and GWAS phenotypes (Supplementary Data 6). To reduce multiple testing for our primary analysis we did not test every GTex tissue; we only tested three relevant esophageal tissue types (esophageal gastroesophageal junction, esophagus mucosa, and esophagus muscularis) as well as whole blood. The total number of genes to test for four tissues is 23,836 (3707, 6944, 6471, and 6714 for each tissue, respectively), resulting in Bonferroni-corrected significance threshold of $2.1 \times 10^{-6}$. Using MetaXcan, we identified three genes (CTD-2228K2.5, CACYBP, and EXOC3) that were not significant in single SNP or MAGMA gene-based testing. We also conducted a secondary analysis examining all 44 GTex tissues, with a more stringent multiple testing threshold to reflect the larger number of tests conducted: in this analysis we identified 5 additional loci not significant in the earlier analysis steps.

**The effect of GERD SNPs on BE/EA.** Since GERD is a risk factor for EA and BE, we investigated whether our significant GERD SNPs were also associated with BE and EA. In practice, many BE or EA cases also have GERD and *just for the purposes of assessing the effect of "GERD only" derived genes on BE/EA* (our main GWAS to determine GERD loci *does* include all GERD cases, including those who have BE and/or EA), we re-ran the GERD GWAS after excluding BE/EA cases and their relatives (*pi-hat >* 0.2). In all, 19 independent significant GERD risk SNPs were identified using GCTA-COJO algorithm[20]. We found 7 of the 19 GERD SNPs were also associated with BE at $P < 0.05$ (binomial probability of this happening by chance $P = 1.8 \times 10^{-6}$), with four associated at Bonferroni-corrected $P < 0.05/19$, and 17 with the same direction of effect (here we considered only the 19 SNPs significant when the GERD GWAS was conducted with BE/EA samples excluded). We found 6 of the 19 GERD SNPs were associated with EA at $P < 0.05$ (binomial probability of this happening by chance $P = 2.3 \times 10^{-5}$), with two at Bonferroni-corrected $P < 0.05/19$, and 17 with the same direction of effect (Supplementary Data 8). In a previous study of BE/EA we identified 14 genome-wide significant SNPs[15]; half of these were associated in our GERD GWAS here, with 2 reaching genome-wide significance (Supplementary Table 9). Although our case numbers were lower for BE/EA compared with GERD, resulting in fewer strongly significant loci for BE/EA (Tables 1 and 2), the GERD-associated SNPs showed good concordance in terms of their estimated effect on BE/EA; the correlations between the estimated log(OR)s for GERD SNPs vs. BE/EA, BE and EA were 0.52 ($P = 4.61 \times 10^{-4}$), 0.42 ($P = 2.65 \times 10^{-3}$), and 0.41 ($P =$

**Table 1 Identified SNPs for GERD (Chromosomes 1–6)**

| Chr | SNP | BP | A1/A2 | OR [95% CI] | GERD adj. P | Meta | BE/EA P | Gene context | eQTL gene | PheWAS |
|---|---|---|---|---|---|---|---|---|---|---|
| 1 | rs1937450 | 66M | T/G | 0.97 [0.96–0.98] | $1.63 \times 10^{-8}$ | --- | 0.73 | [PDE4B]; U4, −81303 | DPYD ($1.9 \times 10^{-7}$) | Obesity ($3.0 \times 10^{-9}$) |
| 1 | rs7552188 | 98M | T/C | 1.04 [1.03–1.06] | $3.76 \times 10^{-10}$ | ++? | 0.54 | [DPYD] | APOB ($2.9 \times 10^{-6}$) | Obesity ($2.1 \times 10^{-10}$) |
| 2 | rs11901649 | 21M | G/A | 1.03 [1.02–1.04] | $2.14 \times 10^{-8}$ | ++? | $1.03 \times 10^{-7}$ | [APOB] | | |
| 2 | rs4362541 | 68M | A/T | 1.05 [1.03–1.06] | $3.38 \times 10^{-10}$ | +++ | 0.0033 | | | Height ($1.9 \times 10^{-5}$) |
| 2 | rs7609078 | 100M | G/A | 1.04 [1.02–1.05] | $2.54 \times 10^{-9}$ | +++ | 0.22 | [AFF3] | | Obesity ($2.0 \times 10^{-7}$), Height ($5.3 \times 10^{-5}$) |
| 3 | rs74652506 | 43M | C/T | 0.96 [0.94–0.97] | $4.16 \times 10^{-8}$ | --- | 0.95 | [ANO10]; SNRK, 51244 | SNRK ($3.2 \times 10^{-8}$) | BMI ($6.9 \times 10^{-33}$), EDU ($3.6 \times 10^{-28}$) |
| 3 | rs7613875 | 50M | C/A | 0.97 [0.96–0.98] | $3.67 \times 10^{-8}$ | --- | 0.068 | RBM6, −5962; MON1A, 4069 | RBM6 ($9.2 \times 10^{-69}$); | Obesity ($5.4 \times 10^{-9}$) |
| 3 | rs4676893 | 71M | A/T | 1.04 [1.03–1.05] | $5.24 \times 10^{-10}$ | +++ | $8.4 \times 10^{-12}$ | FOXP1, −86225 | | Obesity ($7.7 \times 10^{-12}$), smoking ($9.7 \times 10^{-7}$) |
| 4 | rs809955 | 141M | G/A | 1.04 [1.03–1.05] | $4.69 \times 10^{-10}$ | +++ | 0.014 | [MAML3]; BC040304, −174408 | SPIP3 ($2.0 \times 10^{-21}$) | |
| 5 | rs10940767 | 29M | T/A | 0.96 [0.95–0.98] | $3.45 \times 10^{-10}$ | --- | $1.56 \times 10^{-6}$ | AK098570, −142143; LSPIP3, 74104 | | |
| 5 | rs72771256 | 83M | G/A | 1.04 [1.03–1.06] | $2.96 \times 10^{-8}$ | +?+ | 0.046 | HAPLN1, −49454; VCAN, 75804 | | Obesity ($5.3 \times 10^{-8}$) |
| 6 | rs7763910 | 26M | A/G | 0.96 [0.95–0.98] | $2.41 \times 10^{-8}$ | −+ | 0.040 | [LOC285819]; BTN1A1, −28802; BTN2A1, 2789 | BTN1A1 ($5.3 \times 10^{-5}$) | EDU ($4.6 \times 10^{-7}$), Height ($2.8 \times 10^{-6}$) |
| 6 | rs9266237 | 31M | G/C | 0.96 [0.95–0.97] | $1.46 \times 10^{-9}$ | --? | 0.19 | MICA, −42039; HLA-B, 532 | HLA-B ($1.9 \times 10^{-22}$) | |

Chr: (chromosome) and bp: base-pair position of SNP in hg19; A1/A2 refers to the effect allele and noneffect allele for the SNP; OR: GERD meta-analysis odds ratio and 95% confidence interval (confidence intervals were inflated by the square root of the LD-score regression intercept to account for potential residual stratification); GERD adj; P: adjusted P-value of SNP association in meta-analysis of UKBB, i.e., each SNP's chi-squared value was divided by the intercept (1.04) from LD-score regression. 23andMe and QSkin for GERD; The column "Meta" shows the direction of risk association in order of the UKBB, 23andMe and QSkin GERD GWASs where "+" denotes increased risk, and "−" denotes decreased risk, and "?" denotes that the result is missing for that study. P-value BE/EA shows P-value of SNP association for analysis of BE/EA samples. The "Gene context" column refers to the genes nearby (<200 kb) the most associated GERD SNP; brackets indicate that the peak SNP or those in LD ($r^2 > 0.8$) are within the specified genes. The "−" sign after the genes indicates that the genes are located downstream of the top SNP, while the "+" after genes shows that the genes are located downstream of the corresponding SNP. GWAS results for BMI, obesity, education, height and smoking were retrieved from https://genetics.opentargets.org, and are listed in the PheWAS column. If any gene in the column has significant eQTL-association with the SNP from any of the 44 GTEx Tissues (https://gtexportal.org/), the gene is shown in eQTL column and the P-value quoted describes the strength of the relationship to the corresponding SNP. GWAS results for BMI, obesity, major depression, height and smoking were retrieved from https://genetics.opentargets.org, and are listed in the PheWAS column. More detailed eQTL and phewas results are in Supplementary Table 1. For each significant independent association a locuszoom plot is given in Supplementary Fig. 3.

**Table 2 Identified SNPs for GERD (Chromosomes 7–22)**

| Chr | SNP | BP | A1/A2 | OR [95% CI] | GERD adj. P | Meta | BE/EA P | Gene context | eQTL gene | PheWAS |
|---|---|---|---|---|---|---|---|---|---|---|
| 7 | rs4721096 | 1.8M | T/C | 0.96 [0.94–0.97] | $2.24 \times 10^{-8}$ | --- | 0.020 | [MAD1L1]; AK127048, −7624; ELFN1, 89721 | | BMI ($1.1 \times 10^{-12}$) |
| 7 | rs10242223 | 3.5M | A/G | 1.04 [1.02–1.05] | $2.54 \times 10^{-8}$ | +++ | 0.0040 | [SDK1]; DL490859, −28241 | SDK1 ($4.7 \times 10^{-8}$) | Smoking ($2.2 \times 10^{-8}$) |
| 7 | rs10228350 | 114M | A/T | 0.97 [0.96–0.98] | $3.87 \times 10^{-8}$ | --- | 0.024 | [FOXP2] | FOXP2 ($7.7 \times 10^{-8}$) | MDD ($3.3 \times 10^{-9}$) |
| 7 | rs12706746 | 126M | G/A | 0.97 [0.95–0.98] | $1.98 \times 10^{-8}$ | --- | 0.45 | [GRM8] | | MDD ($1.5 \times 10^{-7}$) |
| 11 | rs12792379 | 6M | G/A | 0.96 [0.95–0.98] | $3.91 \times 10^{-8}$ | --- | 0.00023 | CCKBR, −23459; CNGA4, 2839 | CCKBR ($5.0 \times 10^{-12}$) | |
| 12 | rs11171710 | 56M | G/A | 0.97 [0.96–0.98] | $1.80 \times 10^{-8}$ | --- | 0.078 | [RAB5B]; SUOX, −22964; CDK2, 1510 | RAB5B ($1.4 \times 10^{-26}$) | Obesity ($3.3 \times 10^{-9}$) |
| 12 | rs597808 | 112M | A/G | 1.03 [1.02–1.04] | $1.33 \times 10^{-8}$ | +++ | 0.78 | [ATXN2]; BRAP, −106591; SH2B3, 83931 | | Obesity ($1.2 \times 10^{-23}$) |
| 17 | rs34796998 | 50M | C/G | 0.96 [0.95–0.97] | $2.98 \times 10^{-11}$ | --- | 0.073 | CA10, 72144 | | MDD ($7.2 \times 10^{-6}$) |
| 19 | rs1363119 | 18M | A/G | 1.04 [1.03–1.05] | $5.81 \times 10^{-10}$ | +++ | 0.00032 | PGPEP1, −6598; LSM4, 10808 | | |
| 19 | rs12974777 | 19M | C/T | 1.04 [1.02–1.05] | $6.92 \times 10^{-9}$ | +?+ | 0.0014 | [KLHL26]; CRTC1, −2876I; TMEM59L, 36944 | | |
| 21 | rs1297211 | 16M | C/G | 1.03 [1.02–1.04] | $3.64 \times 10^{-8}$ | +++ | 0.31 | NRIP1, −4599 | NRIP1 ($3.2 \times 10^{-5}$) | Obesity ($7.5 \times 10^{-7}$) |
| 21 | rs7282609 | 34M | A/G | 0.97 [0.95–0.98] | $4.18 \times 10^{-8}$ | --- | 0.077 | OLIG2, −119058; C21orf62, 93104 | | EDU ($1.4 \times 10^{-7}$) |

Chr: (chromosome) and bp: base-pair position of SNP in hg19; A1/A2 refers to the effect allele and noneffect allele for the SNP; OR: GERD meta-analysis odds ratio and 95% confidence interval (confidence intervals were inflated by the square root of the LD-score regression intercept to account for potential residual stratification); GERD adj, P: adjusted P-value of SNP association in meta-analysis of UKBB, i.e., each SNP's chi-squared value was divided by the intercept (1.04) from LD-score regression. 23andMe and QSkin for GERD; The column "Meta" shows the direction of risk association for the effect allele in order of the UKBB, 23andMe and QSkin GERD GWASs where "+" denotes increased risk, and "−" denotes decreased risk and "?" denotes that the result is missing for that study. P-value BE/EA shows P-value of SNP association of BE/EA samples. The "Gene context" column refers to the genes nearby (<200 kb) the most associated GERD SNP; brackets indicate that the peak SNP or those in LD ($r^2 > 0.8$) are within the specified genes. The "−" sign after the genes indicates that the genes are located upstream of the top SNP, while the "+" after genes shows that the genes are located downstream of the top SNP. If any gene in the column has significant eQTL-association with the SNP from any of the 44 GTEx Tissues (https://gtexportal.org/), the gene is shown in eQTL column and the P-value quoted describes the strength of the relationship to the corresponding SNP. GWAS results for BMI, obesity, education, major depression, height and smoking were retrieved from https://genetics.opentargets.org, and are listed in the PheWAS column. More detailed eQTL and phewas results are in Supplementary Table 1. For each significant independent association a locuszoom plot is given in Supplementary Fig. 3 (note for rs12974777 the top SNP shown is the top SNP in the region after conditioning on nearby SNP rs1363119).

$3.38 \times 10^{-3}$), respectively (Supplementary Fig. 1c–e). Many of the SNPs in Tables 1 and 2 have EA/BE P-values in the range 0.05 to 1e−4, corresponding to chi-squared variables ranging from 3.84 to 15.13. Since the genome-wide significance threshold ($P = 5e-8$) is 29.7 on the chi-squared scale, for these SNPs we might expect to need BE/EA sample sizes that are between ~2 and ~8 times bigger than are currently available.

**GERD-related traits.** We performed a look-up using the LD hub[21] database to evaluate whether GERD is genetically correlated with other phenotypes. The highest genetic correlations were with education (years of schooling), depression, and BE/EA (Supplementary Data 2). We confirmed the depression result using a recently published larger depression GWAS[22] and obtained a very similar result ($r_g = 0.52$, SE = 0.03). Similarly based on recent GWAS for BMI[23,24], education[24], and height[23], correlation estimates were ($r_g = 0.35$, SE = 0.02), ($r_g = -0.43$, SE = 0.02), and ($r_g = -0.12$, SE = 0.02), respectively (Fig. 3).

**Phenome-wide association.** To further investigate each of GERD-associated SNPs in Tables 1 and 2 against an extensive record of phenotypes, we performed a Phenome-wide association scan (PheWAS) to evaluate the association of our GERD SNPs using the Gene-ATLAS[25] repository (http://geneatlas.roslin.ed.ac.uk/phewas/). Many are associated ($P < 5 \times 10^{-8}$) with a range of complex traits (Supplementary Data 3). In total, 13 of the peak SNPs are strongly associated with BMI or related traits. Two SNPs rs7763910 and rs9266237 are associated with malabsorption/celiac disease. Five of the GERD-associated peak SNPs (rs1937450, rs3106209, rs10242223, rs12706746, and rs967823) are associated with cigarette smoking in Gene-ATLAS.

**Putative drug targets.** We used the online Open-targets drug database (www.targetvalidation.org) to assess if any of the genes implicated in our GERD GWAS are potential drug targets. For each locus in Tables 1–3, we used evidence from eQTL databases, plus gene-based tests in MAGMA and MetaXcan to identify putative target genes of the peak SNPs. We identified seven genes targeted by drugs currently in use or in clinical trials (Table 4). Three of these are existing drug targets for reflux, BE, or esophageal cancer. The remaining four are drug targets for cancer or obesity and may constitute interesting drug targets for reflux and related traits. While we cannot guarantee that the named genes in Table 4 are the correct (or sole) target genes, in each case there is at least some evidence for the named gene. Further details of the drugs used for these genes are in Supplementary Data 4.

## Discussion

Although GERD has been previously established to be heritable, in previous reflux gene-mapping efforts the small effect sizes were an insurmountable problem. In our study, combining across phenotype definitions within UKBB (self-report, ICD10, medication records) and across cohorts was a major factor in our success. For example, a previous GERD GWAS found no genome-wide significant loci[13] and an online convenience analysis of gastroesophageal reflux (gord)/gastric reflux in UKBB ($N = 19,242$ cases, http://geneatlas.roslin.ed.ac.uk/trait/?traits=638) found only two loci (the peak SNPs at these are correlated with the two MHC loci we identify here)—these results have not been published. We identified 25 independent loci in SNP-based tests and a further 23 (20 from MAGMA, 3 from MetaXcan) using gene-based tests.

Several of the genes implicated by our analysis are drug targets, either for drugs already used in GERD, BE, or EA, or for drugs currently used for other conditions. In the latter case, these drugs

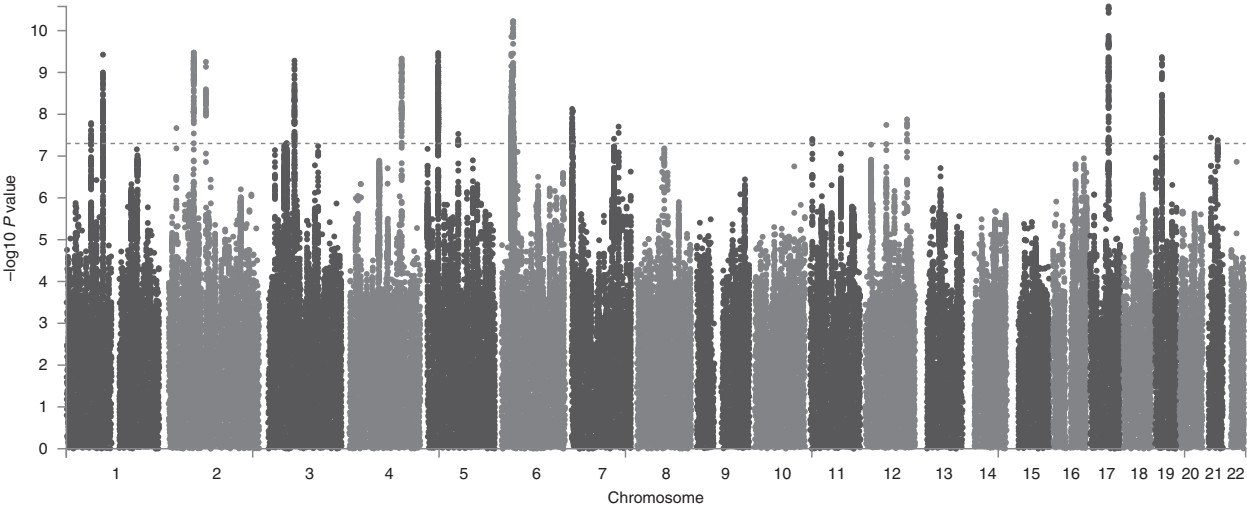

**Fig. 2** Manhattan plot for GERD from meta-analysis of 81,077 GERD cases and 307,284 controls. The x-axis shows genomic position (chromosome 1–22) and the y-axis shows the $\log_{10}$ (P-value) of the SNP association. The threshold for genome-wide significance is set at $P = 5.0 \times 10^{-8}$ (the red-dotted horizontal line)

### Table 3 Additional GERD genes identified via gene-based tests

| Gene | Gene interval | Top SNP | SNP-P | Gene-P | eQTL | PheWAS |
|---|---|---|---|---|---|---|
| *KIAA1522* | 1:3,320,749–33,240,571 | rs61798840 | $2.27 \times 10^{-5}$ | $1.61 \times 10^{-6}$ | | Obesity ($6.5 \times 10^{-6}$) |
| *PDE4B* | 1:66,258,193–66,840.262 | rs1937450 | $1.63 \times 10^{-8}$ | $7.01 \times 10^{-9}$ | | Obesity ($3.0 \times 10^{-9}$) |
| *RABGAP1L* | 1:174,128,552–174,964,445 | rs333423 | $4.70 \times 10^{-7}$ | $6.04 \times 10^{-9}$ | $1.7 \times 10^{-30}$ | MDD ($2.8 \times 10^{-6}$) |
| *MST1R* | 3:49,924,435–49,941,311 | rs7613875 | $4.89 \times 10^{-8}$ | $6.33 \times 10^{-8}$ | $7.9 \times 10^{-12}$ | BMI($6.9 \times 10^{-33}$) EDU($3.6 \times 10^{-28}$) |
| *RBM5* | 3:50126341–50156397 | rs7613875 | $4.89 \times 10^{-8}$ | $5.28 \times 10^{-8}$ | $9.0 \times 10^{-60}$ | BMI($6.9 \times 10^{-33}$) EDU($3.6 \times 10^{-28}$) |
| *SEMA3F* | 3:50192562–50226508 | rs7613875 | $4.89 \times 10^{-8}$ | $3.49 \times 10^{-7}$ | $9.2 \times 10^{-6}$ | BMI($6.9 \times 10^{-33}$) EDU($3.6 \times 10^{-28}$) |
| *MAGI1* | 3:65,339,200–66,024,511 | rs9815340 | $2.36 \times 10^{-6}$ | $2.06 \times 10^{-6}$ | | |
| *LCORL* | 4:17,844,839–18,023,483 | rs1993638 | $1.06 \times 10^{-6}$ | $1.73 \times 10^{-6}$ | | |
| *SNX2* | 5:122,110,691–122,170,234 | rs7707685 | $1.26 \times 10^{-7}$ | $1.30 \times 10^{-6}$ | NA | |
| *SGCD* | 5:155,462,147–156,194,799 | rs56785833 | $2.13 \times 10^{-6}$ | $1.88 \times 10^{-6}$ | $2.4 \times 10^{-18}$ | |
| *LAMA2* | 6:129,204,286–129,837,711 | rs9492232 | $5.97 \times 10^{-7}$ | $2.67 \times 10^{-7}$ | NA | Obesity ($9.4 \times 10^{-9}$), Smoking ($3.4 \times 10^{-6}$) |
| *SNX10* | 7:26,331,515–26,413,949 | rs12700707 | $2.48 \times 10^{-6}$ | $2.32 \times 10^{-7}$ | NA | |
| *VIPR2* | 7:158,820,866–158,937,649 | rs2730263 | $4.95 \times 10^{-7}$ | $1.81 \times 10^{-6}$ | $4.3 \times 10^{-17}$ | Smoking ($4.7 \times 10^{-5}$) |
| *LMX1B* | 9:129,376,722–129,463,311 | rs10760444 | $9.01 \times 10^{-7}$ | $1.87 \times 10^{-7}$ | NA | Obesity ($1.3 \times 10^{-8}$) |
| *ACAD10* | 12:112,212,3857–112,194,911 | rs597808 | $1.33 \times 10^{-8}$ | $2.52 \times 10^{-7}$ | NA | Obesity ($1.3 \times 10^{-23}$) Smoking($1.1 \times 10^{-10}$) |
| *YLPM1* | 14:75,230,025–75,304,013 | rs17183201 | $2.08 \times 10^{-6}$ | $4.98 \times 10^{-7}$ | NA | |
| *SPG7* | 16:89,574,802–89,624,174 | rs57696383 | $2.50 \times 10^{-7}$ | $8.86 \times 10^{-7}$ | $1.4 \times 10^{-21}$ | Height($2.1 \times 10^{-16}$), Obesity ($1.2 \times 10^{-12}$) |
| *SETBP1* | 18:42,258,849–42,648,475 | rs2028653 | $9.73 \times 10^{-6}$ | $1.18 \times 10^{-6}$ | NA | Obesity ($1.6 \times 10^{-4}$), Height ($1.5 \times 10^{-5}$) |
| *DCC* | 18:49,866,542–51,062,273 | rs56796226 | $8.62 \times 10^{-6}$ | $4.98 \times 10^{-10}$ | $2.6 \times 10^{-5}$ | MDD ($1.0 \times 10^{-8}$), EDU ($1.1 \times 10^{-6}$), Height ($1.2 \times 10^{-5}$) |
| *TCF4* | 18:52,889,562–5,330,322 | rs77262239 | $1.02 \times 10^{-6}$ | $8.99 \times 10^{-8}$ | NA | MDD ($4.0 \times 10^{-6}$) |

Gene-based test results from MAGMA software. Regions which already exceeded per-SNP genome-wide significance in Tables 1 and 2 are not shown here. The gene interval in the form Chromosome: start–stop bp is shown mapped to hg19. Top SNP refers to the SNP with the lowest P-value in the region, with SNP-P the P-value for that SNP. Gene-based-P is the gene-based P-value. The eQTL column shows the lowest P-value for testing the correlation between gene expression and genotype of the top SNP in 44 GTEx tissues. PheWAS column shows any strong associations for the top SNP with any of BMI, obesity, education, major depression, height, and smoking. The PheWAS results were retrieved from the online Open-Targets database https://genetics.opentargets.org. They are listed in the PheWAS column alongside its P-value. More detailed eQTL and PheWAS results are in Supplementary Table 14.

should be re-evaluated for possible use with GERD. A subset of the GERD genes also have an effect on BE and/or EA, and are therefore possible drug targets for these conditions: among the putative drug targets in Table 4, the peak SNPs in two genes (*EPHB1* and *CCKBR*) show a larger effect (odds ratio) on BE/EA than they do on GERD. Two further genes (*MST1R* and *CDK2*) show a similar effect size for BE/EA as for GERD (although the P-

values are only $0.05 < P < 0.1$ due to the smaller sample size for BE/EA), whilst three (*PDE4B*, *DYPD*, and *LAMA2*) show no association with BE/EA. In addition to the information in Table 4, *DPYD* has been reported to play a role in chemosensitization in esophageal cancer[26]. For the locus at rs11171710 (chr12:56368078, putative gene *CDK2*) mapping the target gene is difficult as there are many possible target genes in the region.

rs11171710 is an eQTL for multiple nearby genes (*SUOX*, *RPS26*, and *RAB5B*), with *SUOX* also significant in our MetaXcan[19] analysis (Supplementary Data 6). Although there is no eQTL effect on *CDK2*, the peak SNP is 1.5 kb from *CDK2*. *CDK2* is a key cell cycle regulator which inactivates phosphorylation of the RB1 (pRb) tumor suppressor family[27]. Previous work supports the case for the relevance of *CDK2* because proliferation of EA cells is decreased when *CDK2* is downregulated[28].

One of the SNPs (rs11901649, chr2:21250223) that is associated with GERD at genome-wide significance is located in an intron of the *APOB* gene. This variant is strongly associated with

high-cholesterol levels in the UKBB data set (Gene ATLAS $P = 5.26 \times 10^{-89}$), suggesting a potential link between cholesterol levels and GERD risk. A previous observational study also found an association between cholesterol and GERD[29]. This variant was also found to be associated with BE/EA ($P = 1.03 \times 10^{-7}$). The association over *APOB* is 380 kb from a previously reported BE signal[30] over the *GDF7* gene, which also shows some signal for GERD; the peak SNP (rs3072) near *GDF7* has no correlation with rs11901649 ($r^2 = 0.01$), but it has a suggestive level of association with GERD ($P = 1.68 \times 10^{-7}$).

We identified two independent GERD risk loci on chromosome 19, both of which are also associated with BE/EA. The first locus is located near *CRTC1* (rs12974777, chr19:18765663), is an established risk locus for EA[31]. The second locus is located ~400 kb away from *CRTC1* (rs1363119), nearby *GDF15* and *PGPEP1*. Although rs1363119 is not an eQTL in GTEx, tissue and plasma expression levels of *GDF15* associate with BE and EA, with *GDF15* plasma levels influenced by the use of nonsteroidal anti-inflammatory drugs that are known to affect esophageal adenocarcinogenesis[32].

Previous GWASs on EA and BE found genetic associations with rs9257809 (chr6:29356331) in the MHC region[15]. In the present study, we found three independent associations with GERD in this region; rs7763910 (chr6:26472655), rs9266237 (chr6:31325521), and rs114863007 (chr6:34729158). Although the BE/EA SNP rs9257809 and the GERD SNP rs9266237 are 2 Mbp apart in the MHC region, they are in modest LD ($r^2 = 0.12$). SNP rs9266237 showed no association ($P = 0.19$) in our EA/BE dataset. We also found rs9266237 was strongly associated with celiac disease ($P = 1.31 \times 10^{-185}$, Supplementary Data 3).

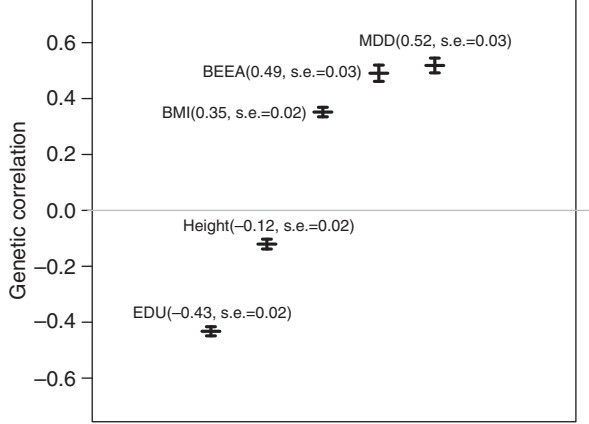

**Fig. 3** Traits with significant genetic correlation with GERD. Vertical axis displays genetic correlation from LD-score regression. Error bars denote ±1 standard error

**Table 4 Putative drug target genes from GERD GWAS**

| Gene | chr | Start, stop | Evidence for this particular gene | Drug(s) | Currently used for GERD/EA/BE | Highly expressed tissues (TPM) |
|------|-----|-------------|-----------------------------------|---------|-------------------------------|-------------------------------|
| *CCKBR* | 11 | 6,280,841–6,293,363 | Peak SNP rs12792379 is an eQTL for *CCKBR* | PROGLUMIDE, NETAZEPIDE | Yes | Stomach (40), brain frontal cortex (22), brain cortex (22), brain cerebellum (22), brain cerebellar hemisphere (22) |
| *EPHB1* | 3 | 134,514,099–134,851,891 | Peak SNP rs60716777 is between *EPHB1* and *KY*; this SNP is eQTL for *KY*, although noteworthy because *EPHB1* is esophageal cancer drug target | VANDETANIB | Yes | Brain-cerebellar hemisphere (33), brain-cerebellum (23), brain-caudate (10), brain-putamen (9.5), cells-EBV (9.3) |
| *CDK2* | 12 | 56,360,556–56,366,568 | See paragraph in discussion section | Multiple | Yes | EBV (64), esophagus-muscularis (32), lung (32), ovary (32), skin (28) |
| *DPYD* | 1 | 97,543,300–98,386,615 | Peak SNP rs7552188 is an eQTL (esophageal mucosa) for *DPYD* | ENILURACIL | No | Transformed fibroblasts (40), lung (27), whole blood (20), spleen (19), ovary (19) |
| *PDE4B* | 1 | 97,543,300–98,386,615 | Signal wholly over *PDE4B*; Significant MAGMA result | Multiple | No | Brain-spinal cord (29), brain-nucleus (17), brain-hypothalamus (14), brain-frontal (14), brain-cortex (14) |
| *MST1R* | 3 | 49,924,436–49,941,306 | Peak SNP rs7613875 is an eQTL (esophagus mucosa) for *MST1R*; significant MetaXcan and MAGMA result | NARNATUMAB, MK-8033, MGCD-265 | No | Brain-spinal cord (30), brain-nucleus (17), brain-substantia (16), brain-frontal (16), spleen (16) |
| *LAMA2* | 6 | 129,204,286–129,837,710 | Signal wholly over *LAMA2*; Significant MAGMA result | OCRIPLASMIN | No | Ovary (121), nerve-tibial (91), fallopian tube (74), transformed fibroblasts (72), cervix-endocervix (55) |

Two further genes (*KCNB2* on chromosome 8, *SLC9A3* on chromosome 5) are drug targets, but we have not listed them in Table 4 as the link from the peak SNP to the putative target gene was weaker than for those in Table 4 (e.g., there are several equally likely candidate genes in the region). The top five expressed genes obtained from GTEx portal (https://gtexportal.org/home/gene, TPM is transcripts per kilobase million).

Several of the top GERD SNPs are associated with traits which are risk factors for GERD (e.g., obesity and smoking). We found strong genetic correlations between GERD and BMI, education, depression, neuroticism, and cigarette smoking. Disentangling the effects of these risk factors is difficult although it is likely that some of these effects are mediated via the GERD risk factor BMI; there are known genetic correlations between education level and BMI[33], while a recent depression study suggested there is a causal link between BMI and depression[22].

This study has some limitations. First, because GERD cases were determined using various sources (ICD10 code, self-reported questionnaires, medical history, and medicine use), the phenotypic definition may not be uniform among all the participating studies. However, the very high-genetic correlation ($r_g >$ 0.9; Fig. 1) between the different GERD phenotypes suggests this is not a major issue. Of particular note, we observed a high genetic correlation ($r_g = 0.94$, SE = 0.018) between GERD phenotypes defined through ICD10 and self-reported medication use, showing that the later can be used as a reliable proxy for ICD10-based GERD diagnosis to increase power. To further confirm that using reflux medicine can be robustly defined as a reflux phenotype, we undertook GWAS on individuals who took reflux medicine but who did not self-report as having reflux, and who do not have an ICD10 medical record of reflux. LD regression was then performed to assess the correlation of this GWAS result with self-reported reflux and ICD10. LD regression analysis indicates a 0.93 (SE = 0.042) and 0.91 (SE = 0.03) correlation with self-reported GERD and ICD10, respectively. The correlation plot between top GWAS results from GERD medicine use with self-reported GERD and ICD10 are shown in Supplementary Fig. 1f, g. The very high genetic correlation of individuals that use reflux medicine with individuals that self-report or who have an ICD10 record of GERD indicates that the use of reflux medication is an appropriate proxy phenotype for classifying an individual as having GERD. Second, although we have attempted to incorporate information on eQTL, for many loci the target gene or genes remains to be determined. While several of the genes highlighted by our GWAS are drug targets, further work will be required to determine if any of the other genes constitute suitable drug targets which can be exploited in the future. Thirdly, although the fact that many of the identified GERD loci are associated with obesity confirms the important role of obesity in GERD risk, when we conducted a formal pathway analysis based on the GERD GWAS meta-analysis, no pathways remained significant following correction for multiple testing (Supplementary Data 7). Finally, although these results may yield putative new drug targets for GERD/BE/EA via repurposing of drugs for other conditions, clearly there is a long way to go from such initial indications to efficacious drug design.

In conclusion, we present here the first successful GWAS reporting genome-wide significant genetic loci for GERD susceptibility. Several of our identified hits are related to established GERD risk factors, BMI, and smoking, with approximately half of them showing associations with BE and EA. Three of the target genes are already GERD/EA/BE drug targets and four others are drug targets for other diseases and as such would be very interesting to investigate for potential medication repurposing for reflux, BE, or EA. Future studies are warranted to further explore the biological significance of these risk loci, and how they may be useful to inform clinical practice and drug development.

## Methods

**UKBB cohort**. UKBB is a cohort study of approximately 500,000 people aged between 40 and 69 years that reside in the UK. All individuals in the UKBB cohort provided informed written consent, and the study was approved by the National Research ethics Service Committee, North West Haydock. All procedures in the research were undertaken in accordance with the World Medical Association Declaration of Helsinki ethical principles for medical research. The Affymetrix UK BiLEVE Axiom array was used to genotype 487,409 participants. Totally, 7.6 million variants with a minor allele frequency (MAF) > 0.01 and HWE $P$-value > $1 \times 10^{-6}$ were successfully imputed. A full description of the UKBB can be found in the report by Bycroft et al.[34]. For this study, we focused solely on 438,870 individuals who were genetically similar to individuals of white-British ancestry based on ancestral principal components (see ref. [35]).

The GERD phenotype data was collated across the following UKBB data fields: self-report (field ID: 20002—Noncancer illness code, self-reported Medical conditions), ICD10 (41202—main diagnoses in ICD10, 41204—secondary diagnosis in ICD10), ICD9 (41203—main diagnoses in ICD9, 41205—secondary diagnosis in ICD9), OPCS (41200—main operative procedures; 41210—secondary operative procedures) (Supplementary Table 1) and treatment/medicine (Supplementary Table 3). Each category was regarded as an indicator of GERD status. The number in each category is summarized in Supplementary Table 4. Individuals who did not have any disorders in their upper digestive system were defined as controls (Supplementary Table 2). In total, there were 68,535 cases and 250,910 controls based on the criteria of a GERD case having at least one of the GERD-positive indicators from above. The average age of the cases is 59.00 (SD = 7.48). The overall average age for UKBB samples is 56.54 (SD = 8.09).

In UKBB BE cases were defined by medical record ICD10 (International Classification of Diseases). The data were extracted from the UKBB Field ID 41202/41204 (ICD10 main and secondary diagnosis) using code K227 (Supplementary Table 5). EA was defined using ICD10 in the UKBB field 40006 (cancer registry) with codes starting with symbol "C15". The cancer tumor histology code in UKBB Field ID 40011 was used to refine the adenocarcinoma cancer type (Supplementary Table 6). The number of BE and EA cases was 2831 and 568, respectively. The average age of the BE cases is 60.42 (SD = 6.60). The average age of the EA cases is 62.43 (SD = 5.43). For the BE and EA analysis, 250,910 controls were selected among the people who did not have any disorders in their upper digestive system.

**23andMe cohort**. As we previously reported[12], 23andMe supplied GWAS summary statistics based on 8743 GERD cases and 43,932 controls of primarily (>97%) European ancestry. All participants provided informed consent under a research protocol that was approved by the AAHRPP-accredited institutional review board, Ethical and Independent Review Services, USA. Genotyped SNPs were filtered using HWE $P$-value > $1 \times 10^{-20}$, MAF > 1%. Cases self-reported whether they have ever been diagnosed by a doctor with heartburn, acid reflux or acid reflux disease, or were treated with medicines for acid reflux/heartburn. Controls were individuals who did not report any symptoms of heartburn, acid reflux, or the use of medications to treat acid reflux.

**QSkin health study cohort**. The QSkin cohort[36] comprises 43,794 participants aged between 40 and 69 years from Queensland, Australia. The work outlined here was approved by the Human Research Ethics Committee of the QIMR Berghofer Medical Research Institute. QSkin participants provided written informed consent to take part in the project. Totally, 17,220 samples were genotyped using Illumina GSA array. GERD cases were defined as individuals who self-reported heartburn and took one or more reflux medications, identified by linkage with the PBS database which captures the use of all prescription medications that are subsidized by the Australian Government (Supplementary Table 7). Individuals whose self-report and medication statuses conflicted were removed. In all there were 2987 GERD cases, together with 10,169 controls (individuals without heartburn).

Genotyped SNPs were filtered using the following criteria; $GenTrain > 0.6$, HWE $P$-value > $1 \times 10^{-6}$, and MAF > 1% using GenomeStudio/BeadStudio and PLINK (version1.9)[37]. In total, 189,387 SNPs failed genotyping quality control leaving 496,695 SNPs for imputation. Samples with >5% missing data were removed. Genotype phasing was performed using Eagle 2[38] and imputation was conducted using minimac version 3[39] through the University of Michigan Imputation Server. SNPs with MAF > 0.01 and imputation quality score >0.3 were taken forward for association analysis.

**Cohort studies for BE and EA**. We obtained GWAS summary statistic results for BE and EA from the following five GWASs of European, North American, and Australian participants[30,40,41]: (1) UKBB; (2) The Barrett's and Esophageal Adenocarcinoma Consortium (BEACON) study; and studies from (3) Bonn; (4) Cambridge; and (5) Oxford. Informed consent was obtained for all participants for all five studies, and ethics approval was obtained from the ethics boards of every participating institution. The total numbers of cases and controls for BE are 8998 and 19,247, respectively. The total numbers of cases and controls for EA are 4680 and 15,751, respectively (Supplementary Table 8). We combined BE and EA as one phenotype (BE/EA) as BE is the premalignant precursor of EA and has a very high-genetic correlation with EA[42].

**Association testing for UKBB cohort**. In UKBB we performed SNP-association testing for GERD using a linear-mixed model implemented in the program BOLT-LMM v2.3[43] to account for cryptic relatedness. Recruitment age, genetic sex, and the first ten principal components were fitted as covariates. We used a sparse set of

360,087 genotyped SNPs spanning the autosomes to derive the Bayesian mixture prior which was subsequently used to model the SNP associations.

Due to the low prevalence of BE/EA which may result in inflated type I error rates in BOLT-LMM[43], a logistic model implemented in PLINK[44] version 1.90b was used for the UKBB BE/EA GWASs. Because the logistic model assumes individuals are unrelated, related individuals were identified based on identity by descent status estimated using autosomal markers, and if two individuals were related (pi-hat > 0.2), one was removed preferentially from the control set. The final number of BE and EA cases becomes 2667 (=2831–164) and 549 (=168–19). The final number of controls for BE and EA becomes 221,787 (=250,910–29,123) and 221,816 (=250,910–29,094), respectively. Sex and recruitment age were fitted as covariates.

**Meta-analysis**. GWAS results for GERD from the UKBB were combined with those from 23andMe and QSkin using a fixed-effects meta-analysis in METAL[45] (2011-03-25 version) using SNP effect sizes and their standard errors. We converted regression coefficients obtained on the quantitative scale from BOLT-LMM into the equivalent log(OR) from logistic regressions for case–control studies using the following formula[46]: $\log(OR) \sim = beta/(mu \times (1 - mu))$, where beta is regression coefficient of the SNP from BOLT–LMM and mu is the proportion of cases in the GWAS. At the completion of the meta-analysis, we used LD-score regression to estimate if there was any inflation due to uncorrected for population stratification[14]. To correct for the slight inflation seen, each SNP's chi-squared value was divided by the intercept (1.04) from LD-score regression results to obtain a final $P$-value

To investigate the association of GERD loci with BE/EA, the BE/EA GWAS results obtained from the UKBB analysis were meta-analyzed with four other datasets[15] using a fixed-effect meta-analysis in METAL[45] (Supplementary Table 8). Since the number of individuals with BE/EA in UKBB was very small, the number of controls was set to four times the number of cases, which were randomly selected from the individuals with no reported (ICD10) upper digestive system problems. To avoid overlapping samples between GERD and BE/EA datasets, we re-ran the GERD GWAS after removing any BE/EA individuals and their relatives (pi-hat > 0.2) from the UKBB GERD dataset.

**Defining independent genome-wide significant SNPs**. We used the stepwise model selection procedure in GCTA-COJO[20] (GCTA software version 1.26) to perform conditional and joint association analysis to identify independent genome-wide significant SNPs. GCTA-COJO uses GWAS summary results, with LD estimated from a reference sample comprising 5000 randomly selected people of white-British ancestry from UKBM. For each index SNP, SNPs within a 10 megabase region (window 10 Mb) were considered for conditioning. We report only SNPs where both the joint and raw $P$-values were $<5 \times 10^{-8}$. The minimum MAF was set at 1%.

**Bivariate LD score regression**. We used LD score regression[14] to quantitatively measure the genetic correlation between traits; this approach takes into account any sample overlap between the input GWASs. We also performed a look-up on the publicly available LD hub[21] database to evaluate whether GERD is genetically correlated to other phenotypes.

**Gene-based tests**. Gene-based tests were conducted with MAGMA[18] based on the per-SNP GWAS summary results for GERD (Supplementary Data 5). We used MAGMA version 1.07 and gene annotations from NCBI Human version 37. We also conducted analysis using MetaXcan[19], a gene-based approach that uses gene expression derived from the GTEx Project data and association summary statistics from GERD GWAS to test the association between genes and GWAS phenotypes (Supplementary Data 6). To reduce multiple testing in our primary analysis we only tested four tissues types; three relevant to GERD (Esophageal Gastroesophageal Junction, Esophagus_Mucosa and Esophagus Muscularis), plus a more generic tissue type with large sample size (whole blood). The total number (23,832) of genes in these 4 tissues was used to determine threshold $P$-value 2.1E−06 (=0.05/23,832) of significant genes. We also conducted a secondary analysis in all 44 GTEx tissues where we corrected for the total number of genes tested across all tissues (Bonferroni significance threshold 0.05/204,388) For all gene-based tests, we used per SNP $P$-value from the GERD GWAS result after correction for the LD-score intercept (1.04).

**Pathway-based tests**. We performed pathway-based enrichment analyses using the GERD meta-analysis results in DEPICT[47]. DEPICT uses the likelihood of involvement of genes in each gene set, based on coregulation of gene-expression data. The preconstituted 14,462 gene sets are used to assess whether candidate genes from the GWAS results are significantly enriched in these gene sets.

**Reporting summary**. Further information on research design is available in the Nature Research Reporting Summary linked to this article.

## Data availability

GWAS summary statistics from the meta-analysis of GERD in UK Biobank and QSKIN can be downloaded from URL (https://doi.org/10.6084/m9.figshare.8986589). GWAS summary statistics for the 23andMe samples are available via direct request to 23andMe (dataset-request@23andMe.com; a data transfer agreement is required). The raw genetic and phenotypic UK Biobank data are available from http://www.ukbiobank.ac.uk/.

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

## Acknowledgements

This work was conducted using the UK Biobank Resource (Application number 25331). This work was supported by funding from the US National Cancer Institute at the National Institutes of Health (Grant number R01CA136725 awarded to T.L.V. and D.C.W.). S.M., R.E.N., and D.C.W. are supported by National Health and Medical Research Council (NHMRC) Fellowships. The QSkin study is supported by Australian NHMRC grant APP1063061. The Swedish Esophageal Cancer Study was funded by grants (R01 CA57947-03) from the National Cancer Institute, and the Swedish Cancer Society (4559-B01-01XAA; and 4758-B02-01XAB). The Kaiser Permanente Study was supported by US NIH grants R01DK63616 and R01CA59636) and from the California Tobacco Related Research Program (3RT-0122 and 10RT-0251). The MD Anderson controls were drawn from dbGaP (study accession: phs000187.v1.p1) following approval. Genotyping of these controls (C. Amos, PI) were performed through the University of Texas MD Anderson Cancer Center (UTMDACC) and the Center for Inherited Disease Research (CIDR), supported in part by NIH grants R01CA100264, P30CA016672, and R01CA133996, the UTMDACC NIH SPORE in Melanoma 2P50CA093459, as well as by the Marit Peterson Fund for Melanoma Research. CIDR is supported by contract HHSN268200782096C. The UK Biobank was established by the Wellcome Trust medical charity, Medical Research Council (UK), Department of Health (UK), Scottish Government, and Northwest Regional Development Agency. It also had funding from the Welsh Assembly Government, British Heart Foundation, and Diabetes UK. We thank Scott Wood, John Pearson, and Scott Gordon from QIMR Berghofer for support. We thank the research participants and employees of 23andMe who contributed to this study.

## Author contributions

J.A. and S.M. conceived the research project. J.A., P.G., M.H.L., J.S.O., X.H., and S.M. analyzed the data. J.A., P.G., M.H.L., J.S.O., X.H., C.M.O., R.E.N., J.L., T.L.V., R.T., I.G., A.B., J.J., R.C.F., J.S., C.P., D.C.W., and S.M. contributed to the data collection and contributed to genotyping. J.A., P.G., J.S.O., and S.M. wrote the first draft of the paper. All authors contributed to the final version of the paper.

## Additional information

**Competing interests:** The authors declare no competing interests.

## BEACON

Marilie D. Gammon[15], Douglas A. Corley[15], Nicholas J. Shaheen[15], Nigel C. Bird[15], Laura J. Hardie[15], Liam J. Murray[15], Brian J. Reid[15], Wong-Ho Chow[15], Harvey A. Risch[15], Weimin Ye[15], Geoffrey Liu[15], Yvonne Romero[15], Leslie Bernstein[15] & Anna H. Wu[15]

[15]International Barrett's and Esophageal Adenocarcinoma Consortium, Mountain View, CA, USA

## 23andMe Research Team

M. Agee[16], B. Alipanahi[16], A. Auton[16], R.K. Bell[16], K. Bryc[16], S.L. Elson[16], P. Fontanillas[16], N.A. Furlotte[16], D.A. Hinds[16], K.E. Huber[16], A. Kleinman[16], N.K. Litterman[16], M.H. McIntyre[16], J.L. Mountain[16], E.S. Noblin[16], C.A.M. Northover[16], S.J. Pitts[16], J. Fah Sathirapongsasuti[16], O.V. Sazonova[16], J.F. Shelton[16], S. Shringarpure[16], C. Tian[16], J.Y. Tung[16], V. Vacic[16] & C.H. Wilson[16]

[16]23andMe, Inc., Mountain View, CA 94041, USA

