## [Peer Review File · Nature Communications]

Reviewers' Comments:

Reviewer #1:

Remarks to the Author:

I reviewed the previous version the authors submitted to NG. In general the authors address my concerns well.

I only have one minor comment.

Since the phenotype definition for the three cohorts is slightly different, it would be helpful to present QQ plots of UKBB, QSKIN and 23andMe separately. Current manuscript only includes the meta-analysis QQ plot.

Reviewer #3:

Remarks to the Author:

Overall, the paper is well written and useful.

Dr. MacGregor and his colleagues reported a large meta-GWAS analysis of GERD, followed by integrative analyses with BE/EA GWAS and other external information, such as eQTL and drug targets. The results they reported here are new and of general interest to the community. The majority of concerns from previous reviewers has been addressed satisfactorily. There are just a few concerns and clarifications below that I suggest the authors may take into consideration.

By saying "risk factor" in the paper, did the authors mean causality? If that, I wonder whether GERD is really a strong risk factor for BE/EA (Line 55-56), or just happens to be correlated due to unknown cofounders (both genetic and environmental). Similarly, obesity and smoking are risk factors for GERD (Line 320). Does a Mendelian Randomization analysis help in this case? If previous studies already provided enough evidence on these, please cite.

Did the authors check the expression patterns of GERD genes across tissues and cell types? Any of them show tissue-specific expression pattern? In the future study, it could be of interest to integrate the GWAS with GTEx to infer the relevant tissues for GERD, instead just arbitrarily using Esophageal Gastroesophageal Junction, Esophagus Mucosa and Esophagus Muscularis as the relevant tissues to conduct MetaXcan analysis.

Although GWAS for self-reported medication and ICD defined GERD have been conducted separately, and their genetic correlation was as high as 0.91 (SE = 0.03), I am still a bit worry that the long-term taking medicine could be a confounder between self-reported medication and ICD defined GERD (who may also take medicine), leading to the observed high genetic correlation. It could be hard to solve this using the current data set if all the cases take medicine for a long term.

For the drug target part, it could be better to check which tissue/cell types the putative target genes are highly and specifically expressed in. It should be noted that there is a long way to go from GWAS to drug design.

Line 55-56: The authors may add some references for this strong statement, i.e., "Gastroesophageal reflux disease (GERD), the frequent regurgitation of stomach acid and bile, is the main risk factor for both BE and EA"

Table 1&2: For eQTL-genes, why not show their relevant tissues? Normally, tissue-specific cis-eQTLs are more biologically meaningful than the common ones. MAF of these SNPs may also be of importance to show, although I know this will make Tables too big.

Line 76: How about "identifying genetic variants for GERD" instead of "identifying genetic for

GERD”.

Line 137 (footnote, Table 1): at least delete the third “allele”

Line 206: please replace “sex” with “6”

Line 213-217: The authors found 6-7 out of 19 GERD SNPs had P values ranging from 0.05 to $1e-4$ for BE/EA. How likely this could be observed just by chance? How about the performance of top SNPs of BE/EA on GERD?

Line 349: please delete “the” at the end of this line

Methods: Please clarify what threshold was used for SNP QC in each cohort, to ensure reproducibility. Did the authors use the same threshold across all the cohorts (like Line 426-433)?

Reviewers' comments:

Reviewer #1 (Remarks to the Author):

I reviewed the previous version the authors submitted to NG. In general the authors address my concerns well.

I only have one minor comment.

Since the phenotype definition for the three cohorts is slightly different, it would be helpful to present QQ plots of UKBB, QSKIN and 23andMe separately. Current manuscript only includes the meta-analysis QQ plot.

Response:

We have added these QQ plots for UKBB, 23andMe and QSKIN into supplementary figure 2.

Reviewer #3 (Remarks to the Author):

Overall, the paper is well written and useful.

Dr. MacGregor and his colleagues reported a large meta-GWAS analysis of GERD, followed by integrative analyses with BE/EA GWAS and other external information, such as eQTL and drug targets. The results they reported here are new and of general interest to the community. The majority of concerns from previous reviewers has been addressed satisfactorily. There are just a few concerns and clarifications below that I suggest the authors may take into consideration.

By saying "risk factor" in the paper, did the authors mean causality? If that, I wonder whether GERD is really a strong risk factor for BE/EA (Line 55-56), or just happens to be correlated due to unknown confounders (both genetic and environmental). Similarly, obesity and smoking are risk factors for GERD (Line 320). Does a Mendelian Randomization analysis help in this case? If previous studies already provided enough evidence on these, please cite.

Response:

High quality epidemiological studies demonstrate consistently that GERD is a strong risk factor for BA/EA (Gharahkhani et al. 2016; Coleman, Xie, and Lagergren 2018) , and that risk rises with duration and severity of symptoms. We have added a reference to the Coleman paper to the introduction to reinforce this. The strong genetic correlations we found between GERD and BE/EA suggest a clear genetic link, although inferring causality for these (plus obesity, smoking as the reviewer suggests) is not a trivial undertaking. Such a causal analysis requires a detailed Mendelian Randomization analysis, including work on multiple risk factors considered simultaneously (likely a mediation or multivariate mendelian randomization analysis). This work would be a whole other manuscript in and of itself.

Did the authors check the expression patterns of GERD genes across tissues and cell types? Any of them show tissue-specific expression pattern? In the future study, it could be of interest to integrate the GWAS with GTEx to infer the relevant tissues for GERD, instead just arbitrarily using Esophageal Gastroesophageal Junction, Esophagus Mucosa and Esophagus Muscularis as the relevant tissues to conduct MetaXcan analysis.

Response:

We have now run the MetaXcan analysis for all 44 GTEx tissues and present the results in sup table 15. For our primary analysis, to reduce multiple testing, we focus on the 4 tissues most likely to be of interest (Esophagus_Gastroesophageal_Junction, Esophagus_Mucosa, Esophagus_Muscularis plus Whole Blood). For this 4 tissue analysis, we report the p-value from MetaXcan after Bonferroni correction by the number of genes in the 4 tissues (pvalue divided by #genes in the 4 tissues --23,957).

We have added a secondary analysis looking at all 44 GTEx tissues. For this we correct for all of tissues tested (Bonferroni correction for 204,388); when this is done we identified 5 additional loci where the P-value was smaller than $2.45E-07(=0.05/204388)$. We have included the 5 loci as an additional sheet in sup table 15 (selected_genes(44_tissues)), with details of which tissue gave the most significant result. We have also included the full MetaXcan results for all tissues as additional sheets in sup table 15. We have also added this in the Results section of our manuscript.

Although GWAS for self-reported medication and ICD defined GERD have been conducted separately, and their genetic correlation was as high as 0.91 (SE = 0.03), I am still a bit worry that the long-term taking medicine could be a confounder between self-reported medication and ICD defined GERD (who may also take medicine), leading to the observed high genetic correlation. It could be hard to solve this using the current data set if all the cases take medicine for a long term.

Response:

To address this we examined the correlation between UKB self-reported medication defined GERD and GERD from one of the other independent cohorts (23andMe). The genetic correlation between GWAS result from UKB medication only and self-reported GERD in 23andMe was very high ($r_g=1.1775$, $SE=0.1756$), suggesting that there is very good concordance between “medication defined” GERD and self reported GERD. Note that LD-Score does not restrict the estimates of correlation to be between -1 and +1; in this case by chance the estimate drifts a non-significant amount above 1. In any event, it is clear that the genetic correlation is likely to be close to 1.

For the drug target part, it could be better to check which tissue/cell types the putative target genes are highly and specifically expressed in. It should be noted that there is a long way to go from GWAS to drug design.

Response:

We agree with the reviewer that there is a long way to go from GWAS to drug design. We have added the following sentence to our limitation section in the discussion:

“although these results may yield putative new drug targets for GERD/BE/EA via repurposing of drugs for other conditions, clearly there is a long way to go from such initial indications to efficacious drug design.”

We have added a column in Table 3 to show the top 5 tissues in which that gene is most highly expressed.

Line 55-56: The authors may add some references for this strong statement, i.e., “Gastroesophageal reflux disease (GERD), the frequent regurgitation of stomach acid and bile, is the main risk factor for both BE and EA”

Response:

We have added additional references; Koek et al, Cook et al and Coleman et al.

Table 1&2: For eQTL-genes, why not show their relevant tissues? Normally, tissue-specific cis-eQTLs are more biologically meaningful than the common ones. MAF of these SNPs may also be of importance to show, although I know this will make Tables too big.

Response:

We have now added in the relevant tissues in column “eQTL(gene and tissue)” in sup table 1 and in column “eQTL(tissue)” in sup table14 as the reviewer suggests. We all added MAF in sup table1 and sup table 14. To prevent the tables becoming too big, we have not added these in the main manuscript.

Line 76: How about “identifying genetic variants for GERD” instead of “identifying genetic for GERD”.

Response:

We have made this change.

Line 137 (footnote, Table 1): at least delete the third “allele”

Response:

We have made this change.

Line 206: please replace “sex” with “6”

Response:

We have made this change.

Line 213-217: The authors found 6-7 out of 19 GERD SNPs had P values ranging from 0.05 to 1e-4 for BE/EA. How likely this could be observed just by chance? How about the performance of top SNPs of BE/EA on GERD?

Response:

The probability of 6 out of 19 SNPs having a $P < 0.05$ if there were no true associations is $P = 2.3e-5$; that is, this is unlikely to have happened by chance. The analogous probability for 7 of 19 SNPs is $P = 1.8e-6$.

We have added a sup table (Supplementary Table 19), which shows the GERD results in 14 top BEEA SNPs in our previous study (Gharahkhani, Fitzgerald, et al. 2016). 2 SNPs have $P < 5.0E-8$ whilst 5 further SNPs have $P < 0.05$.

Line 349: please delete “the” at the end of this line

Response:

We have made this change.

Methods: Please clarify what threshold was used for SNP QC in each cohort, to ensure reproducibility. Did the authors use the same threshold across all the cohorts (like Line 426-433)?

Response:

23andMe kindly provided us with GWAS summary statistics. In 23andMe's standard QC pipeline they included the following filters: Hardy-Weinberg $P > 10^{-20}$ and MAF $> 1\%$. We also used UKBB genotype data which was QC'd prior to our analysis. From the UKBB documents, the main QC criteria were Hardy-Weinberg $P > 10^{-6}$ and MAF $> 1\%$. This information has been added to the manuscript.

Reviewers' Comments:

Reviewer #3:

Remarks to the Author:

The authors addressed my concerns well. I do not have any other comments.